# Sustainable environmental remediation via biomimetic multifunctional lignocellulosic nano-framework

Jinghao Li[1,2,3], Xiaohan Li[1,2], Yabin Da[4], Jiali Yu[1,2], Bin Long [1,2], Peng Zhang[1,2], Christopher Bakker[3], Bruce A. McCarl[4], Joshua S. Yuan[5] & Susie Y. Dai [1,2✉]

Chemical pollution threatens human health and ecosystem sustainability. Persistent organic pollutants (POPs) like per- and polyfluoroalkyl substances (PFAS) are expensive to clean up once emitted. Innovative and synergistic strategies are urgently needed, yet process integration and cost-effectiveness remain challenging. An in-situ PFAS remediation system is developed to employ a plant-derived biomimetic nano-framework to achieve highly efficient adsorption and subsequent fungal biotransformation synergistically. The multiple component framework is presented as Renewable Artificial Plant for In-situ Microbial Environmental Remediation (RAPIMER). RAPIMER exhibits high adsorption capacity for the PFAS compounds and diverse adsorption capability toward co-contaminants. Subsequently, RAPIMER provides the substrates and contaminants for in situ bioremediation via fungus *Irpex lacteus* and promotes PFAS detoxification. RAPIMER arises from cheap lignocellulosic sources, enabling a broader impact on sustainability and a means for low-cost pollutant remediation.

[1] Synthetic and Systems Biology Innovation Hub, Texas A&M University, College Station, TX 77843, USA. [2] Department of Plant Pathology and Microbiology, Texas A&M University, College Station, TX 77843, USA. [3] Department of Chemical Engineering, Texas A&M University, College Station, TX 77843, USA. [4] Department of Agricultural Economics, Texas A&M University, College Station, TX 77843, USA. [5] Department of Energy, Environmental, and Chemical Engineering, Washington University in St. Louis, St. Louis, MO 63130, USA. ✉email: sydai@tamu.edu

Pollution from persistent organic chemicals (POPs) seriously threatens human and ecosystem health[1]. Environmental remediation of POPs presents a global challenge[2,3] and often involves expensive, complicated multi-step processes[4]. To date, remediation approaches often involve a treatment train encompassing pollutant adsorption[5], detoxification, and subsequent material degradation[6]. However, the existing practices are labor-intensive, costly, often disjointed and generating secondary pollution[7]. Thus it is critical to develop synergized strategies that are well-integrated, cost efficient, environmentally benign, sustainable, and effective[8,9].

Efficient and integrated remediation solutions are particularly needed for remediating the common and widespread POP forms such as per- and polyfluoroalkyl substances (PFAS)[10]. PFAS is highly recalcitrant to degradation, causes severe damage to human and wildlife health, and is pervasive including being detected in remote areas like the Arctic ocean[11]. In humans, PFAS has been found to increase cancer, birth defects and incidence of compromised immune systems[12–14]. There are over 9000 alternative PFAS molecules and the U.S. EPA has recognized the risk and is developing drinking water regulations related to two PFAS forms, perfluorooctanoic acid (PFOA) and perfluorooctanesulfonic acid (PFOS)[15]. Currently, the only commercial PFAS remediation method involves costly and unsustainable thermal deconstruction[16]. Even though many sorbents have been explored for environmental remediation[17–21], their use is hindered by the need of high-cost metals or polymer-based materials and the creation of secondary pollution[7,22]. While more sustainable and lower-cost chitosan, biochar, microbial biomass, and agricultural waste-based biosorbents have been studied, they lack the performance required for industrial applications[23]. Bioremediation has been explored for PFAS remediation, but is limited by low efficiency, slow processing times, and inability to remove trace level contaminants[24,25]. It is thus critical to address the challenges in sorbent, bioremediation, and treatment train integration with innovative, sustainable, efficient, integrative, and cost-effective solutions.

Here we show a bioinspired system design to address the aforementioned challenges by exploiting renewable engineered nanomaterials to sequentially adsorb pollutions, store them for bioremediation, host the microbes to degrade the toxics to a more benign form, and finally degrade themselves. The nano-structural and functional designs give rise to the Renewable Artificial Plant for In-Situ Microbial Environmental Remediation (RAPIMER) from chemically modified lignocellulosic biomass. RAPIMER overcomes the challenges in sorbent, bioremediation, and treatment train integration with several important characteristics. First, it is engineered from low-cost and widely available cellulose and lignin in a way that exploits their hydrophilic and hydrophobic properties to create an amphiphilic environment for highly efficient PFAS adsorption. Second, RAPIMER provides a natural substrate for bacterial and fungal growth, on which the white rot fungus decomposes the PFAS and the later the RAPIMER itself synergistically without creating secondary pollutants. In particular, RAPIMER can present the toxic substrates at much higher bioavailability to improve efficiency and trace-level PFAS remediation, uniquely integrating the treatment train. Third, with lignin as a natural and recalcitrant substrate, RAPIMER supports the expression of redox enzymes to degrade PFAS into benign substances. RAPIMER thus is a single plant-based nanomaterial that enables the whole treatment train within itself. As shown below, it exhibits high capacities of adsorption, empowers efficient toxics degradation and ultimate source material degradation, plus is made from low-cost materials. Thus, it presents a cost-effective and sustainable way to remediate PFAS and more generally POPs.

## Results

**RAPIMER component design for adsorption and biodegradation.** As aforementioned, RAPIMER-based remediation relies on a material design that: (1) efficiently adsorbs POPs/PFAS, (2) holds the PFAS at high concentration, (3) serves as substrate for white rot fungus that carries out bioremediation, and (4) stimulates the expression of biodegradation enzymes for rapid and efficient degradation (Fig. 1 and Supplementary Fig. 1). To achieve this, multifunctional composites were designed to integrate natural biopolymers lignin and cellulose in a form that achieves effective adsorption and remediation (Supplementary Fig. 1). The natural biopolymers ensure seamless integration of adsorption with bioremediation, as the RAPIMER could serve the substrate that hosts white rot fungus for biodegradation[26,27] (Fig. 1). Meanwhile, lignin and cellulose were sourced from corn stover, a widely available agricultural residue (see Methods for detailed procedures).

In the cellulose component design, the TEMPO-oxidation processing was used to derive cellulose nanofibrils with significantly finer diameter compared to the source cellulose fibers (Supplementary Fig. 2). Furthermore, compared with cellulose fiber, the cellulose nanofibrils exhibited much better hydrostability (Supplementary Fig. 3) and a larger specific surface area (Supplementary Table 1). Cellulose nanofibrils were therefore chosen as a component to build the RAPIMER nano-framework. As for the lignin component, the purified lignin from a residual solution of cellulose manufacturing showed no binding affinity for PFAS (Supplementary Table 2). Thus, to increase the affinity and considering the positive charge of PFAS, the lignin was modified by grafting it with polyethylenimine (PEI) (Fig. 1a), as the polyethylenimine is biocompatible and can introduce cationic interactions to enhance adsorption[28]. This modification resulted in positively charged lignin particles[29] that enhanced adsorption by interacting with negatively charged PFAS molecules.

The chemically modified lignin and cellulose nanofibrils were then fabricated into a stable RAPIMER composite using freeze drying and oven curing (Fig. 1c) (see Methods for detailed procedures). The resultant RAPIMER possesses a three-dimensional (3D) nano-structure (Fig. 2) with a large surface area for PFAS adsorption, the positively charged lignin that attracted the PFAS, and spatial accommodation for biodegrading microorganisms. Subsequently, analysis by kinetic adsorption and retention tests (Fig. 1d) revealed that RAPIMER exhibited high adsorption capacity and efficiency. (See Methods for details on procedures). Further biodegradation assay revealed that RAPIMER can serve as a substrate for fungal growth (Fig. 1e) and promote the biotransformation of the stored PFAS into two less toxic shorter chain products ($C_7HF_{13}O$ and $C_6HF_{11}O_2$)[30].

**3D structural and multifunctional design of RAPIMER.** As stated above, the 3D structural design was crucial for efficient PFAS adsorption. The RAPIMER had a finer nanoscale fiber structure compared with that of cellulose fibers (Fig. 2a, b, and Supplementary Fig. 2) and formed a spatial lattice structure for efficient PFAS adsorption. The average fiber diameter for the resultant cellulose nanofibrils is 2.35 nm, as compared to the original 11.57 μm of cellulose fiber (Fig. 2c). This resulted in a 13fold increase in specific surface area (Supplementary Table 1).

The 3D nano-structure of RAPIMER was compared with other composites (Fig. 2d–i). Particularly, the cellulose fibers possessed a limited number of unconfined hydroxyl groups, whereas the cellulose nanofibrils in RAPIMER formed a flake-like 3D lattice scaffold due to the abundant hydrogen bonding among the well-distributed nanofibrils (Fig. 2g). In contrast, the

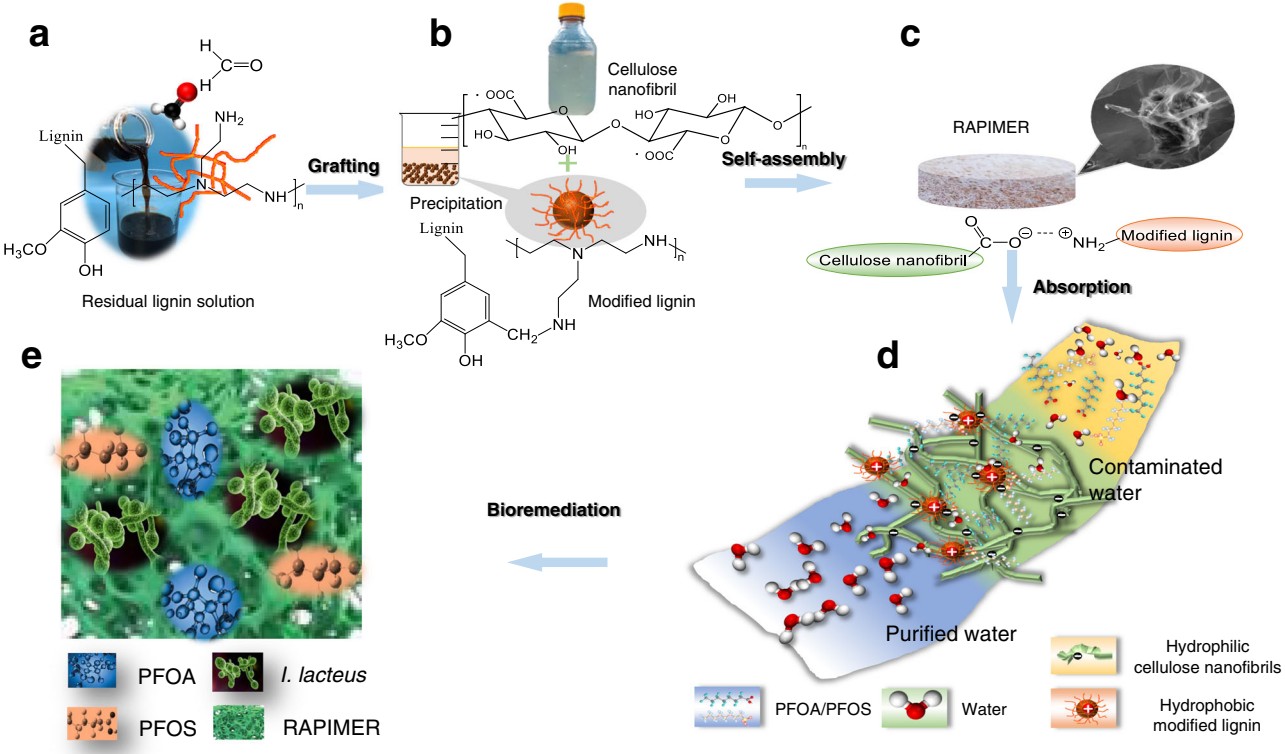

**Fig. 1 The design strategy, fabrication process, chemical adsorption, and fungus degradation scheme of the RAPIMER system. a** Corn stover residual lignin solution and selective graft reaction using formaldehyde and polyethylenimine to produce positively charged modified lignin particles. **b** Corn stover derived cellulose nanofibrils prepared by TEMPO-oxidation method and modified lignin chemical structure. **c** The modified lignin and nanocellulose nanofibrils formed RAPIMER composite through self-assembly by the formation of carboxylic acid/amine interaction. **d** PFAS adsorption by the RAPIMER composite. **e** Fungal bioremediation through co-metabolism and biodegradation of PFAS and RAPIMER system.

cellulose fibers possess a tube-like structure. The nano-structure of RAPIMER enlarged surface area, enabled a much better hydrostability, and provided a hydrostable framework for high-capacity PFAS adsorption.

Leveraging a reverse engineering principle, chemically modified lignin was imbedded in cellulose nanofibril foam to form the RAPIMER for efficient adsorption and bioremediation integration as a biomimetic substrate. Since lignin is hydrophobic, the polyethylenimine-modified lignin retains its hydrophobicity. Supplementary Fig. 4a, b shows the typical SEM images of lignin and modified lignin, which had similar morphologies. However, the images from Energy-Dispersive X-ray (EDX) mapping highlighted that the modified lignin had increased nitrogen content compared to that in the unmodified lignin (increased from 0.2 wt% to 24.2 wt%). The FTIR was also employed to characterize the structural change of lignin after chemical modification. The unmodified lignin spectrum presented a band at about $3400\,cm^{-1}$ (the hydroxyl groups), a band at $1650\,cm^{-1}$ (conjugated C–O vibration), bands at $1598\,cm^{-1}$, $1510\,cm^{-1}$ and $1423\,cm^{-1}$ (the aromatic vibrations), bands around $1460\,cm^{-1}$ (C–H bending), and the bands at $1218\,cm^{-1}$ and $1120\,cm^{-1}$ (stretch on the C–O linkage). After chemically grafting with polyethylenimine, the modified lignin spectrum showed the amine peak at about $3380\,cm^{-1}$, the C–H stretching, and scissoring bands at $2940–2830\,cm^{-1}$ and $1463\,cm^{-1}$, which were consistent with the polyethylenimine spectrum. The most significant change after chemical modification was the appearance of the peaks around $1656\,cm^{-1}$ and $1599\,cm^{-1}$, which were assigned to the amide I and amide II with O = C stretch. These bands indicated that the polyethylenimine was successfully grafted onto the aromatic ring of the lignin structure, introducing

abundant amine groups on the modified lignin particles. The successful chemical modification reaction was further confirmed by the XPS analysis (Fig. 2k and Supplementary Fig. 5). A signal at about 400 eV corresponding to N1s was identified in the modified lignin spectrum. The polyethylenimine grafting thus resulted in abundant positively charged amine groups on lignin and RAPIMER, which could adsorb negatively charged molecules like PFAS.

The further analysis of FTIR spectra of the cellulose nanofibrils, cellulose fiber/lignin composite, and the RAPIMER composite (cellulose nanofibrils/modified lignin composite) were shown in Supplementary Fig. 6. In the FTIR spectra, the COOH peak at $1720\,cm^{-1}$ (in both cellulose nanofibrils and cellulose nanofibrils/lignin composite spectra) completely disappeared in the RAPIMER spectrum. The results highlighted that the positively charged amine groups in the modified lignin reacted with the negatively charged carboxyl groups from cellulose nanofibrils. As a result, the peak at $1720\,cm^{-1}$ of cellulose nanofibrils shifted to $1600\,cm^{-1}$ in the RAPIMER spectrum. This peak transformation can be attributed to the formation of carboxylic acid/amine salt[31]. Thus, the cellulose nanofibril C6-carboxyl groups provided the anchoring sites to integrate with the modified lignin amine groups and formed the stable 3D composite structure (Fig. 1c). Due to these chemical and structural changes, RAPIMER also showed excellent thermal stability and would not decompose until the temperature reached 200 °C (Supplementary Fig. 7). In summary, the 3D nano-structure for RAPIMER was unique in that the negatively charged cellulose nanofibrils (hydrophilic) and the positively charged lignin (hydrophobic) generated an amphiphilic environment and a 3D lattice structure with a large surface area for highly efficient PFAS adsorption.

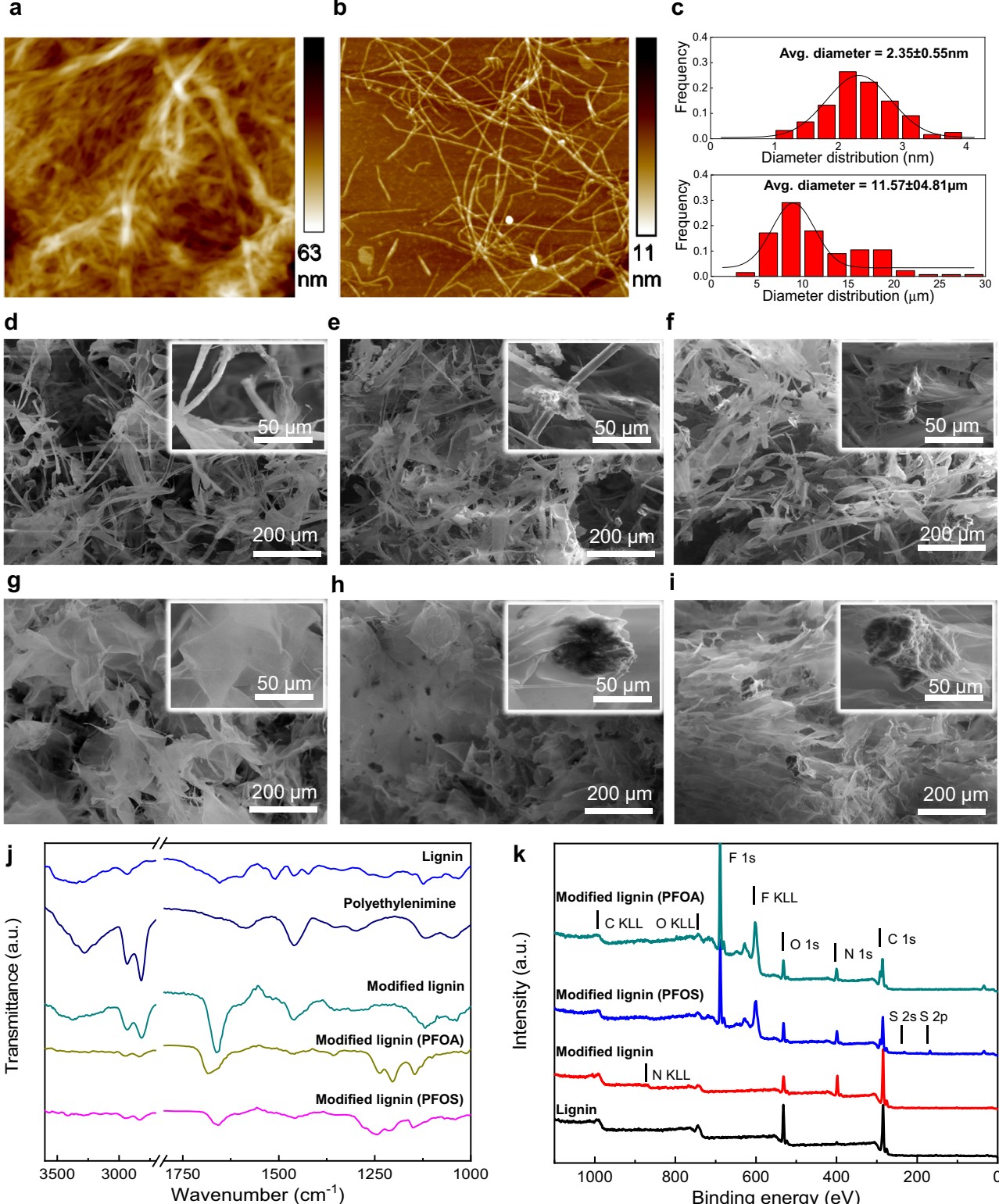

**Fig. 2 The characterization of the bioinspired composite materials and their components. a** The AFM image of cellulose fibers. **b** The AFM image of cellulose nanofibrils. **c**, The average diameter estimates of cellulose fibers and cellulose nanofibrils. **d** The typical SEM image of cellulose fibers, **e** The typical SEM image of cellulose fiber/lignin composite. **f** The typical SEM image of cellulose fiber/modified lignin composite. **g** The typical SEM image of cellulose nanofibrils. **h** The typical SEM image of cellulose nanofibril/lignin composite. **i** The typical SEM image of RAPIMER composite. **j** The FTIR spectra of lignin, polyethylenimine, modified lignin, and modified lignin after PFAS adsorption. The top light blue line: lignin, the second top dark lune line: polyethylenimine, the third green line: modified lignin, the fourth yellow-green line: PFOA adsorbed modified lignin, the bottom purple line: PFOS adsorbed modified lignin. **k** The XPS spectra of different lignin, modified lignin, and modified lignin after PFAS adsorption. The top green line: PFOA adsorbed modified lignin. The second blue line: PFOS adsorbed modified lignin. The third red line: modified lignin. The bottom black line: lignin. For SEM images, the experiment was reproduced $n = 3$ times. The Source data is provided as a Source Data file.

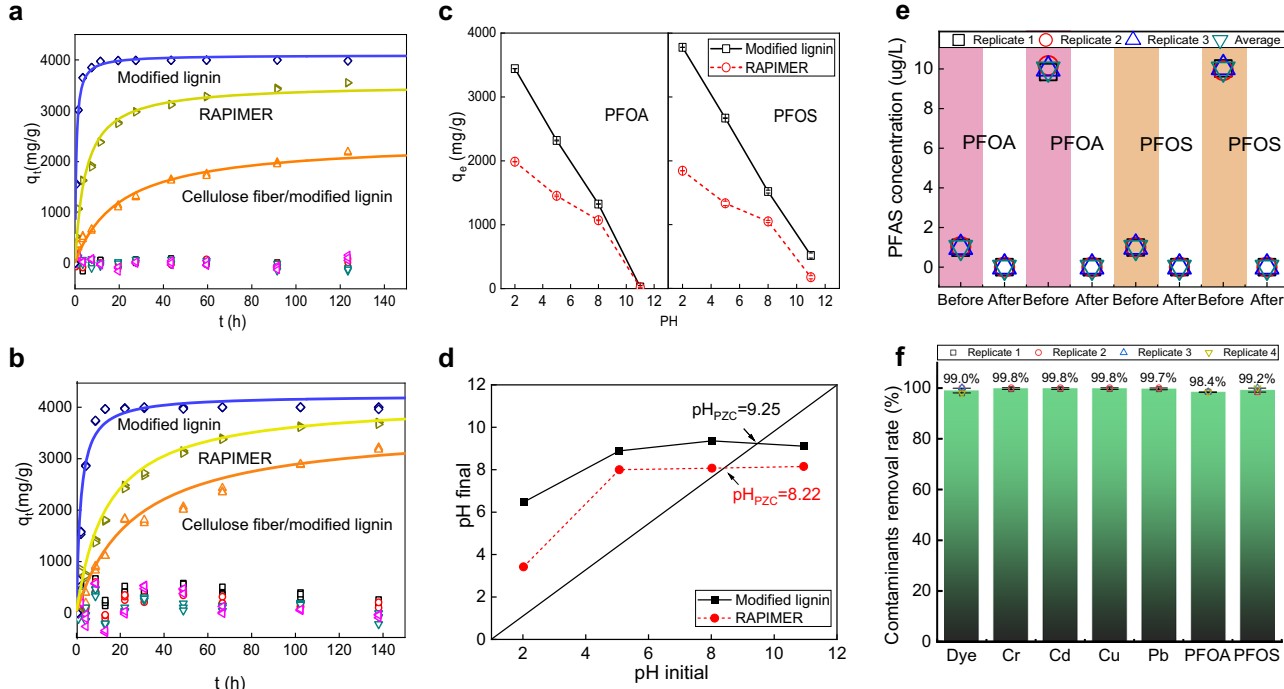

**Fig. 3 The characterization of material/PFAS adsorption. a** The PFOA adsorption kinetics of the bioinspired composite and their components. **b** The PFOS adsorption kinetics of the bioinspired composite and their components. Blue diamond and curve: modified lignin, yellow-green triangle and curve: RAPIMER, orange triangle and curve: cellulose fiber/modified lignin composite, green triangle: cellulose nanofibrils, purple triangle: cellulose nanofibril/ lignin composite, green square: cellulose fibers. Each time points are triplicate measurements. Due to the small variations, some replicates are overlapped and all points are shown in the figures (the invisible standard derivations are not applied in the figure due to the small variations). **c** pH dependence of the modified lignin and RAPIMER composite adsorption capacity. Red open circle, RAPIMER composite; black open square, modified lignin. **d** pH of zero charge of the modified lignin and RAPIMER composite. Red solid circle, RAPIMER composite; black solid square, modified lignin. **e** Adsorption efficiency of the RAPIMER composites for PFOA and PFOS at 1 and 10 µg/L. The peristaltic pump filtration system was employed for this PFAS adsorption measurements (Supplementary Fig. S8). The experiment was reproduced $n = 3$ times. **f** Adsorption efficiency of the RAPIMER composites for 1 mg/L PFOS, PFOA, anionic dye, chromium (Cr), cadmium (Cd), copper (Cu), lead (Pb) mixtures in the rain water. The peristaltic pump filtration system was employed for this PFAS adsorption measurements (Supplementary Fig. S8). The mean with error bar represents the average value and standard deviation (SD) of experimental replicates. The experiment was reproduced $n = 4$ times (The ICP-MS measurement was reproduced twice due to the high precision). The Source data is provided as a Source Data file.

**High adsorption efficiency for PFAS and co-contaminant removal.** One important feature is the RAPIMER adsorption capacity, which was examined using adsorption kinetics analysis. The RAPIMER composite reached PFOA and PFOS adsorption equilibrium within 30 h and 45 h, respectively (Fig. 3a, b, yellow-green triangle). The adsorption quantity per gram sorbent ranged between 3529 mg/g and 4151 mg/g (Supplementary Table 2), which is among the largest reported adsorption among the various sorbents in the literature (Supplementary Table 3).

Component adsorption experiments showed that modified lignin was the primary constituent material that adsorbed both PFOA and PFOS (blue diamond) with cellulose not playing a role (Fig. 3a, b, green square and triangle). Experiments with different compositions showed a 1:1 ratio of polyethylenimine-grafted lignin to cellulose nanofibrils in RAPIMER yielded the largest adsorption capacity. The negatively charged cellulose nanofibrils could have provided the 3D structural framework to immobilize the PFAS after binding. Adjusted by the sole lignin weight, the RAPIMER lignin component showed a 71.9% and 95% increase in adsorption capacity for PFOA and PFOS compared to modified lignin particles, which exceeded performance by any type of PFAS sorbents in previous studies (Supplementary Table 3).

The RAPIMER adsorption was pH-dependent, where the PFAS adsorption decreases in the pH range smaller than eight (Fig. 3c and Supplementary note 3). However, the point of zero charge of RAPIMER was 8.22, which warranted broad applications in water treatment (Fig. 3d). The RAPIMER 3D nano-structure design thus rendered both higher PFAS adsorption capacity and more adaptation to pH variations. The RAPIMER adsorption of PFAS was tested in flowing solutions containing mixed PFAS molecules (1:1 PFOA and PFOS by weight). The RAPIMER material was packed into an online filter in a flow system (Supplementary Fig. 8) and subject to filtering a PFAS solution at 1 µg/L concentration. The concentration is lower than many of the environmental relevant conditions, where PFAS are detected up to 20 µg/L or even higher concentrations[32]. The lower concentration is to sufficiently test the absorption capacity at low concentration. Resultant adsorption was more than 99.90% for PFOA and 100% for PFOS (Fig. 3e), indicating RAPIMER has great potential to remove trace-level POPs[33]. The trace level PFAS removal uniquely enables the integration with bioremediation.

Furthermore, RAPIMER was tested for water treatment under more realistic conditions in the presence of potential co-contaminants of PFAS. Rain water was collected, filtered, and spiked with a mixture of PFAS, heavy metals, and another organic pollutant. Figure 3f shows the high removal efficiency of PFAS together with five other contaminants (i.e., an anionic dye molecule, chromium (Cr), cadmium (Cd), copper (Cu), and lead (Pb) at 1 mg/L for each contaminant in the rain water) when using the same flow system (Supplementary Fig. 8). The result

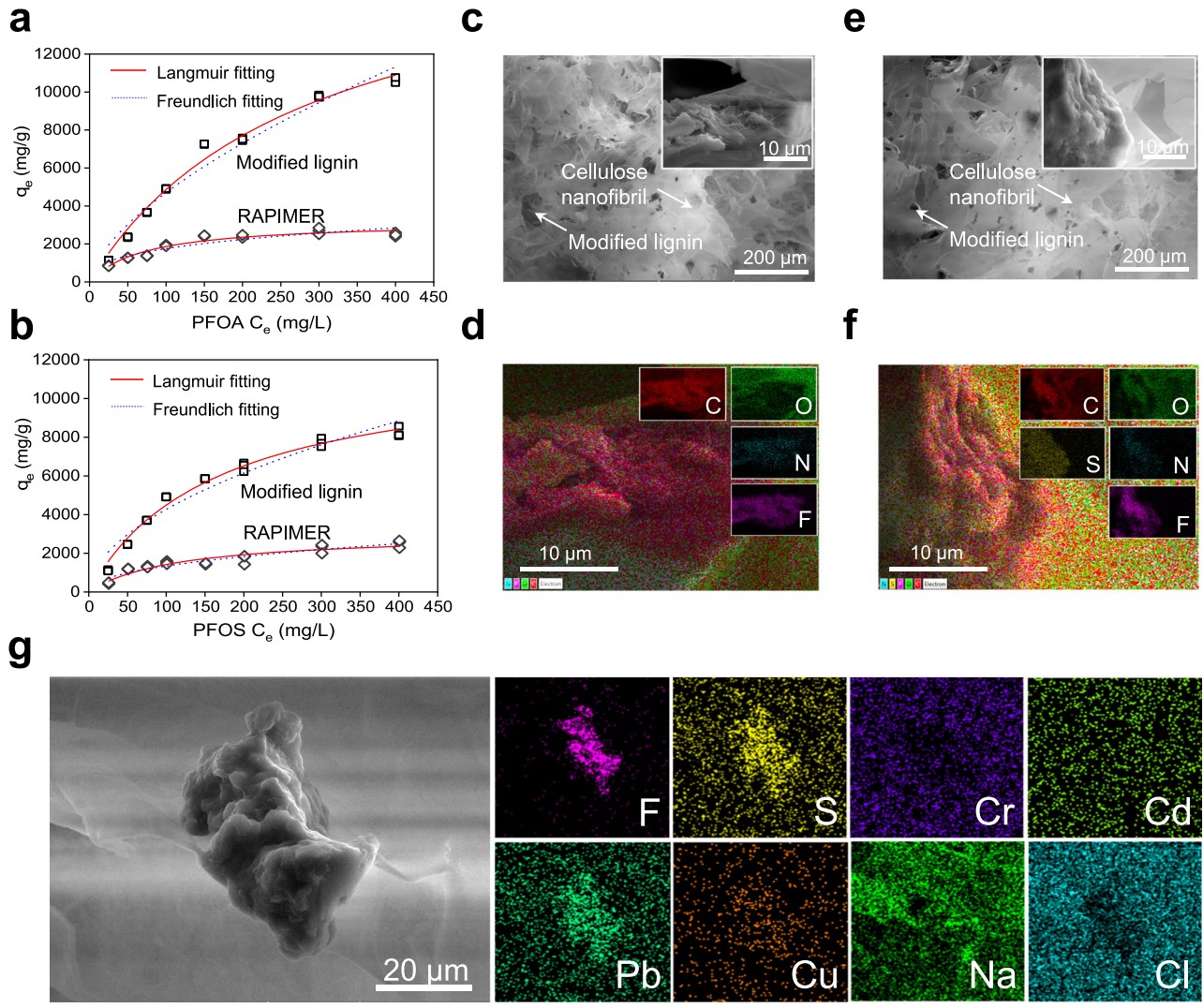

**Fig. 4 The isothermal models and absorption mechanism. a** The adsorption isotherms of the modified lignin and RAPIMER composite binding with PFOA. **b** The adsorption isotherms of the modified lignin and RAPIMER composite binding with PFOS. The solid red line: Langmuir fitting model. The dotted blue line: Freundlich fitting model. Top fitting: modified lignin. Bottom fitting: RAPIMER. Each concentration had triplicate measurements and all overlapped points were shown in the figures (the invisible standard derivations are not applied in the figure due to the small variations). The experiment was reproduced $n = 3$ times. **c** The SEM image of PFOA adsorbed RAPIMER composite. **d** The EDX image of PFOA adsorbed RAPIMER composite with color coding for different elements. Red: carbon, green: oxygen (28.0 wt%), blue: nitrogen (1.4 wt%), purple: fluorine (21.4 wt%). **e** The SEM image of PFOS adsorbed RAPIMER composite. **f** The EDX image of PFOS adsorbed RAPIMER composite with color coding for different elements. Red: carbon (51.0 wt%), green: oxygen (27.2 wt%), blue: nitrogen(1.1 wt%), purple: fluorine(19.0 wt%), yellow: sulfur (1.7 wt%). **g** The SEM and EDX images of PFAS/dye/metals contaminants as-adsorbed RAPIMER composite with color coding for different elements. purple: fluorine, yellow: sulfur, blue violet: chromium, light green: cadmium, emerald green: lead, orange: copper, green: sodium, and blue green: chlorine. The results were based on three repeated measurements and all replicates are shown in the figures. The experiment was reproduced n = 3 times.

suggested that around 99% of all the pollutants were adsorbed by the RAPIMER packed filter (Fig. 3f).

**Mechanisms for the high adsorption capacity of RAPIMER and co-contaminant removal**. Both computational modeling and material characterization were carried out to reveal mechanisms for the high RAPIMER adsorption capacity. Using data from an adsorption isotherms test (Fig. 4a, b), the Langmuir and the Freundlich models were fit with the results listed in Supplementary Table 4. The Langmuir model showed a better data fit according to the resultant $R^2$ (Supplementary Table 4). This suggests that the adsorption likely happened on the monomolecular layer on the RAPIMER surface due to the combined electrical charge attraction and hydrophobic interaction between PFAS and the modified lignin. Specifically, the carboxyl or

sulfuric group of PFAS is negatively charged and hydrophilic, whereas the C-F chain is more hydrophobic. Therefore, the positively charged amino groups on the modified lignin provided more unconstrained sites to attract negatively charged PFAS molecules with electrostatic force. Furthermore, the modified lignin retained hydrophobicity from its lignin precursor. The hydrophobicity allows the interaction with PFAS C-F chain to further stabilize PFAS adsorption. The negatively charge of cellulose nanofibrils with a spatial lattice structure provided independent building blocks that stored the PFAS molecule adsorbed to modified lignin, which prevent the release of PFAS adsorbed to modified lignin and thus improve the total adsorption capacity. Thus the positive charge introduced by polyethylenimine modification, the negative charge of the cellulose nanofibrils, and the hydrophobic backbone of lignin all contributed to a 3D

amphiphilic porous nano-framework to facilitate PFAS adsorption. The unique RAPIMER design has enabled significantly increased PFAS adsorption as compared to previously reported sorbents (Supplementary Table 3).

SEM (Fig. 4c, e) and EDX (Fig. 4d, f) analyses further confirmed the aforementioned adsorption mechanisms. Nitrogen and fluorine (N, blue color; F, purple color, see Fig. 4d, f) were introduced from the grafting polyethylenimine and PFAS adsorption, and could serve as probes for lignin and PFAS localization. As shown in Fig. 4d, f, the blue-purple overlapping region suggests that the modified lignin (nitrogen from polyethylenimine) interacted with PFAS (fluorine from PFAS) in a result consistent with the isothermal modeling. Thus, the high adsorption capacity of RAPIMER results from the universal distribution of positively charged modified lignin particles in the 3D hydrophobic and negatively charged nano-framework.

As for the adsorption of co-contaminants, the morphology and chemical element analysis (Fig. 4g) indicated that the 3D dual-electrical/amphiphilic structural design of the RAPIMER provided independent building blocks to adsorb PFAS, metals, and organic dye molecule. The SEM image showed the central CSML particle is surrounded by the CSCNF network. Moreover, the EDX images showed that the fluorine and metals were trapped by different RAPIMER components (CSML and CSCNF). The fluorine distribution image (Fig. 4g) indicated PFAS were adsorbed primarily onto CSML as shown in the image center, considering that PFAS are the only chemicals containing fluorine. The adsorption of the anionic organic dye and positively charged metal ions are more universally distributed in RAPIMER. The EDX analysis demonstrated the diverse adsorption capacity could be attributed to the 3D dual-electrical/amphiphilic structure design, which had the potential to remove PFAS in a complex matrix and remediate a large range of environmental contaminants under realistic conditions.

FTIR and XPS analyses provided additional information on the PFAS adsorption mechanism. The FTIR analysis (Fig. 2j) showed the spectra changes in the modified lignin upon adsorbing PFOA and PFOS. The amine band (around $3380 \, cm^{-1}$) disappeared, and the major peaks of $-CF_2$, $-CF_3$ occurred around $1238 \, cm^{-1}$, $1205 \, cm^{-1}$, and $1150 \, cm^{-1}$. Meanwhile, peaks of $-COO^-$ ($1681 cm^{-1}$, for PFOA) and $-SO_3^-$ ($1215 \, cm^{-1}$, for PFOS) were observed in the lignin spectra after PFOA and PFOS adsorption. The analysis thus further confirmed that the amine groups in the modified lignin were the primary functional groups to capture PFAS by the electrostatic interactions. The XPS spectra of modified lignin before and after PFAS adsorption also confirmed the interaction between the materials and PFAS (Fig. 2k and S7). The primary peak at $689 \, eV$ (F1s peaks in PFAS adsorbed modified lignin spectra) suggested a stable and robust interaction between the modified lignin and PFAS. Overall, the results highlighted the effectiveness of the nano-structural and chemical design of RAPIMER to create a unique amphiphilic nano-porous composite material that can bind PFAS with electrostatic interaction and further stabilize this binding by the hydrophobic interaction and negatively charged environment. The large surface area of the cellulose nanofibril framework further improved the adsorption capacity to achieve the highest reported adsorption capacity. We then evaluate if RAPIMER can be consumed by microorganisms to integrate bioremediation.

**RAPIMER sustained fungal growth**. RAPIMER was found to provide essential nutrients to maintain fungal growth, allowing the integration of adsorption and biodegradation as a sustainable treatment train approach. White rot fungus *Irpex lateus* has been widely used in bioremediation[34] and it can efficiently degrade

POP compounds (organic dyes) in the presence of lignin, which serves as a natural recalcitrant substrate to induce the redox enzymes for efficient bioremediation[26,27]. Figure 5a, b shows fungal growth as measured by a fungal viability assay based on fungal protein quantitation (see Methods for details of the fungal viability assay). Among the tested bioinspired composites, *I. lacteus* grew best on the RAPIMER composite at a 1:1 ratio composition. Less *I. lacteus* mycelium was found on the cellulose nanofibril/lignin composite, followed by cellulose nanofibrils during the two-week evaluation period (Fig. 5a, c). After adjusting the polyethylenimine content, the optimized degrading RAPIMER composite had a cellulose nanofibril to lignin ratio at 5:3, which is consistent with levels found in natural plant cell walls[35,36]. The results highlighted the effectiveness of the reverse engineering principle in manufacturing RAPIMER as the biomimetic natural substrates.

Fungal growth conditions were further evaluated on RAPIMER composites treated with PFAS at various concentration levels (10, 100, 1000, and 10,000 μg/L, see Methods for detailed procedures). *I. lacteus* grew well on RAPIMER treated with all four concentrations with a better performance on the intermediate PFAS concentrations (100 and 1000 μg/L—Fig. 5b) as shown by fungal viability assay. Digital microscopy also revealed the fungus hypha positioned well on the RAPIMER material surface (Fig. 5d). Overall, RAPIMER was found to be effective at sustaining fungal cell growth and presenting concentrated PFAS to fungal bioremediation.

**PFAS degradation in RAPIMER**. To be useful in remediation, the RAPIMER needs to promote PFAS degradation. The fungal growth on PFAS-concentrated RAPIMER suggested the degradation capability. High-resolution mass spectrometry analysis was employed to quantitatively measure PFAS-derived products in the growing solution. Two biotransformed products were identified ($C_7HF_{13}O$ and $C_6HF_{11}O_2$) as a result of degradation in the two-week evaluation period. The two compounds could be perfluoroheptanal (C7) and perfluorohexanoic acid (PFHA, C6) (M/Z at 346.9839 and 312.9723, individually). Previous studies by Mahapatra et al and others have shown that the shorter chain PFBA (C4), PHPA (C5), and PFHA (C6) are much less toxic than the PFOA (C8), suggesting the degraded products detected in this study are potentially less toxic than PFOS and PFOA[30,37]. These results highlighted the feasibility using RAPIMER to integrate the treatment train via adsorption and subsequent bioremediation to treat heavily PFAS-contaminated media.

**Molecular mechanisms causing RAPIMER-induced PFAS biodegradation**. We carried out systems biology proteomics analysis to reveal the molecular mechanisms behind PFAS biodegradation (See methods). Proteomics analysis was performed on four different fungal growth conditions (i.e., *I. lacteus* grew on cellulose nanofibrils, cellulose nanofibrils/lignin composite, RAPIMER, and PFAS enriched RAPIMER). Previous results suggested that lignin is a recalcitrant natural substrate that enhances fungal organic azo dye degradation due to overexpression of redox enzyme network[26,27]. The results above suggested that lignin can synergize fungal biotransformation yet PFAS also induces the expression of detoxification enzymes. Figure 5e lists the top-ranked differentially expressed proteins in the PFAS enriched RAPIMER compared to RAPIMER alone. We found that protein categories hydrolase, dehydrogenase, oxidoreductase, thioredoxin, reductases, and defense-related enzymes were overexpressed in the PFAS enriched RAPIMER and exhibited several features. First, the overexpression or presence of defense-related enzyme cytochromes P450 in the PFAS treatment condition suggested that cytochrome

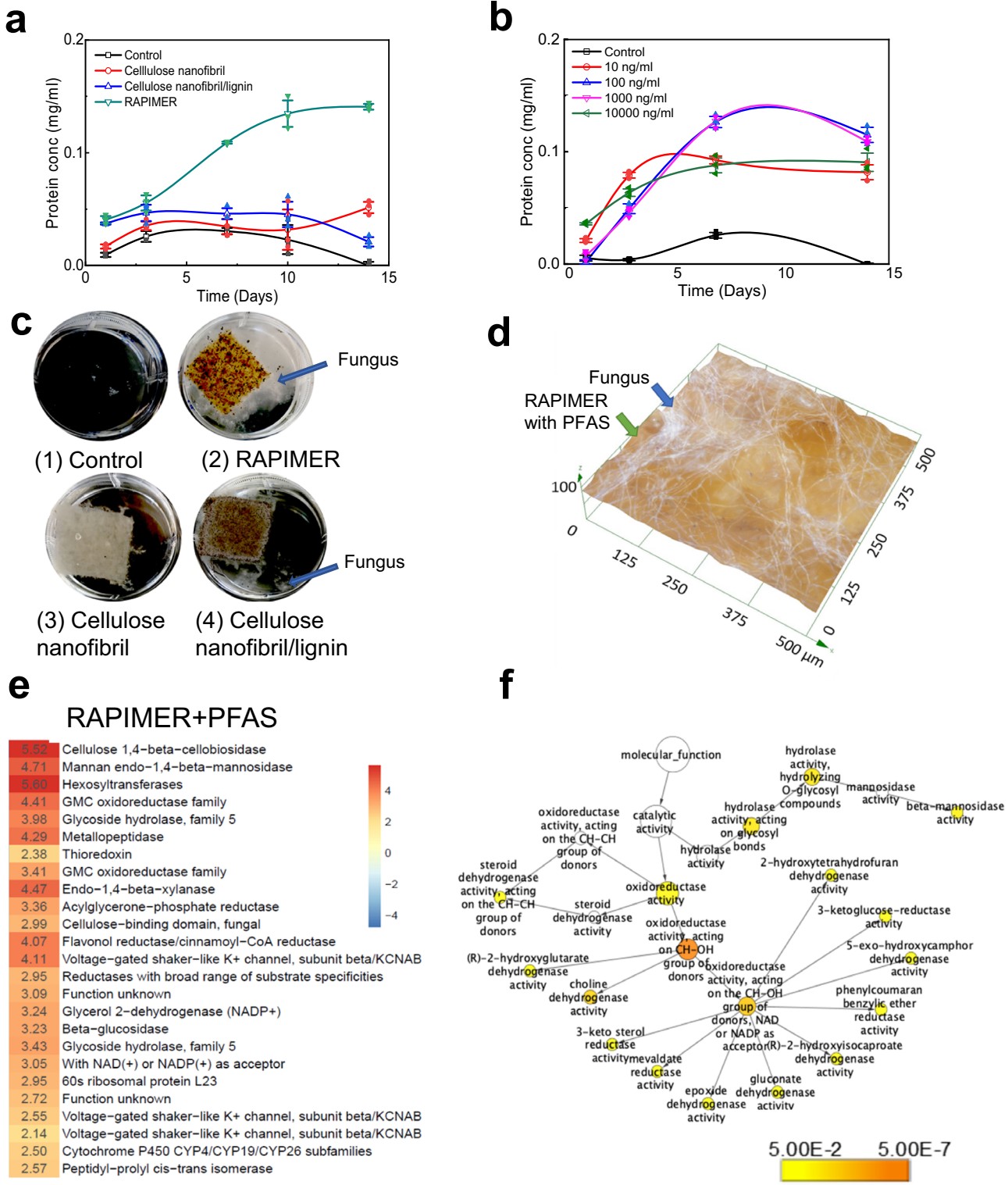

**Fig. 5 The assessment of fungal degradation of RAPIMER integrated system. a** The fungal growth curve on different bioinspired composites. Top green line: RAPIMER composite, blue line: cellulose nanofibril/lignin composite, red line: cellulose nanofibril, black line: glucose deprived Kirk media as the control. **b** The fungal growth curve on RAPIMER composite treated with solutions in different PFAS concentrations. The blue line: 100 μg/L. The purple line: 1000 μg/L. The green line: 10,000 μg/L. The red line: 10 μg/L. The black line: Kirk media as the control. All PFAS solution is the mixture of PFOA/PFOS (1:1). For a and b, the experiment was reproduced $n = 3$ times. The mean with error bar represents the average value and standard deviation (SD) of experimental replicates. **c** The microscope images of fungal growth on different materials. (1) Glucose-derived Kirk media, (2) RAPIMER composite, (3) Cellulose nanofibril composite, (4) cellulose nanofibril/lignin composite. The background is adjusted into black to show the fungus mycelia. **d** The digital microscopy image of *l. lacteus* hyphae growing on the RAPIMER substrate adsorbed with PFAS. **e** The RAPIMER-induced overexpressed enzymes upon PFAS treatment. The numbers denoted the normalized spectra counts for the proteins in the heat map. **f** The GO enriched pathway analysis of upregulated proteins in the PFAS treatment conditions.

P450 enzymes could be involved in PFAS transformation in fungus. P450 enzymes were known for detoxification functions and bioremediation potentials, and was also suggested for potential role in PFAS transformation[38]. Second, the general category of oxidoreductase was upregulated. In fact, gene ontology analysis (Fig. 5f and Supplementary excel sheet 1) highlighted that two types of oxidoreductases (NADPH/NADH-dependent or not) were differential regulated. Some of these oxidoreductases synergize with cytochromes and catalyzes the monooxygenase reaction ($RH + O_2 + 2e^- + 2H^+ \rightarrow ROH + H_2O$) by forming the CH-OH group, assisting the PFAS biotransformation reactions. Third, the overexpression of redox enzymes and detection of C7 and C6 metabolites is consistent with the previously reported "one carbon ($-CF_2-$) removal pathways"[39]. This overexpression suggested that the fungal cells adapted to the oxidative stress allowing the RAPIMER composite to achieve a higher degradation capacity[40]. Fourth, the cellulose and xylose deconstruction enzymes were significantly overexpressed in the PFAS-containing RAPIMER. The cells might need to mobilize more carbon source for biodegradation. More importantly, the results highlighted that the RAPIMER can be degraded by fungus efficiently. Overall, the systems biology analysis highlighted that RAPIMER synergized with the fungal growth to induce the redox biocatalytic network, in turn promoting PFAS degradation. RAPIMER also provided a solid substrate to allow the fungus to adapt to a highly PFAS-concentrated environment uniquely enabling the integration of treatment train.

**Environmental benefits of RAPIMER remediation**. The environmental implications of RAPIMER use were investigated using a multiple dimensional life cycle assessment (LCA) in comparison with other commercially utilized PFAS sorbents (see Methods and Supplementary Table 5). The PFAS sorbents we considered involved activated carbon and ion exchange resins[41]. Our LCA analysis examined acidification, greenhouse gas emissions, human toxicity (cancer and non-cancer), ecotoxicity, ozone depletion, particulate matter, and surface ozone formation. Specifically, we compared the results in a normalized manner; that is, we examined the environmental impacts in treating 1 $m^3$ PFAS-contaminated groundwater (with a concentration rate of 0.21 μg/L) using each of the sorbents (note that a comparison based on 1 kg of produced sorbent was also performed, see Supplementary Fig. 9). In doing the normalization, we factored in sorbents' different adsorption capacities (Supplementary note 6). For instance, we found the RAPIMER PFAS adsorption capacity is about 3-10 times that of the activated carbon[42,43] and 1-3 times that of the ion exchange resins (Supplementary Table 6). The normalized comparison revealed RAPIMER significantly reduced net $CO_2$ emissions relative to activated carbon (exhibiting 1.4e-7 kg $CO_2$ per 1 $m^3$ groundwater when treated with RAPIMER versus 4.5e-6 kg with activated carbon) and a moderate emission reduction compared with ion exchange resin (3.4e-7 kg). Figure 6 also presents results for the other environmental categories in comparison with those for activated carbon and ion exchange resins. Compared with activated carbon, the results in Fig. 6 show that RAPIMER use greatly lowers environmental impacts for all environmental impact's categories. The RAPIMER use also significantly outperforms exchange resins for human cancer toxicity and ozone depletion. To further investigate this, a contribution analysis was performed and revealed the needed chemicals and production of cellulose nanofibrils component were the main contributors to the RAPIMER environmental impacts (i.e., accounting for over 70% in most of the categories). In contrast, the energy input (i.e., electricity usage) played a minor role, for instance, accounting for 3.9% of the total greenhouse gas global warning potential.

RAPIMER degradability, PFAS degradation, and lack of secondary pollutions lead to additional environmental benefits that were not included in this LCA analysis[44]. Use of less energy, and no significant PFAS reemission are also benefits relative to existing practices[45]. In addition, its self-degradation alleviates the need for additional equipment and additional associated environmental burdens.

## Discussion

The study designs a renewable biomimetic composite material, RAPIMER, which uniquely adsorbs PFAS, holds it, and presents it for degradation to achieve a whole remediation treatment train in one plant-based material. Also, compared to other sorbents, RAPIMER achieves unprecedented adsorption capacity for PFAS and broad adsorption capability toward other co-contaminants. It is derived from inexpensive lignin, and cellulose source, which is sustainable and environmentally friendly. It is a thermostable, hydrostable, biodegradable, renewable, and low-cost multifunctional material that integrates the entire treatment train. Additionally, RAPIMER, compared to other environmental consequences from conventional PFAS treatments involving sorbents or combustion, brings significant environmental and sustainability benefits as compared to the environmental consequences from conventional PFAS treatments involving combustion.

Thus RAPIMER holds excellent potential for PFAS bioremediation and other co-contaminant removal. The biomimetic material does not need scarce and expensive chemicals and has high efficiency and a large remediation capacity. It overcomes the low bioremediation efficiency, sustains microbial growth, and works in situ without introducing secondary hazards. RAPIMER also provides additional detoxification mechanisms through inducing the redox enzyme overexpression by biomimetic substrate and sustaining microbial cell growth with high concentration of toxics. The nanomaterial design also enables future advancement. We fabricate RAPIMIER from cellulose and lignin from corn stover to optimize cellulose fiber oxidation and surface area while maximizing ionic interaction, which enhances stability, adsorption, and bioremediation. The hydrophilic yet negatively charged cellulose nanofibrils and the hydrophobic positively charged modified lignin create a unique amphiphilic and dual charge environment reducing PFAS desorption and maximizing adsorption efficiency. Overall, RAPIMER enables a unique environmentally benign and efficient PFAS remediation platform. Beyond these unique strengths, the reverse engineering principle, treatment train integration, and co-substrate molecular mechanism employed to make RAPIMER hold out possibilities for broader remediation applications. The principle and mechanisms can be broadly adapted to remediate numerous other hazardous chemicals.

## Methods

**Materials**. Corn stover was obtained from a cornfield after harvest in College Station, TX. Sodium hydroxide (NaOH), ammonium acetate, ethylenediaminetetraacetic acid (EDTA), sodium dodecyl sulfate, magnesium chloride hexahydrate, 37% formaldehyde solution, polyethylenimine (PEI), citric acid, Bradford reagent, urea, thiourea, trizma base, 3-(4-Heptyl)phenyl-3-hydroxypropyl) dimethylammoniopropanesulfonate, direct red 81, 95% perfluorooctanoic acid (PFOA), and 98% purity perfluoro-1-octanesulfonic acid (PFOS) were purchased from Sigma Aldrich. 1,4 Dithiothreitol was purchased from Roche Diagnostics. (2,2,6,6-Tetramethylpiperidin-1-yl) oxyl (TEMPO), sodium bromide (NaBr), and sodium hypochlorite (NaClO) were purchased from Fisher Scientific. The isotope-labeled PFOA (Perfluoro-n-[1,2-13C2] octanoic acid and sodium perfluoro-1-[1,2,3,4-13C4] octanesulfonate were purchased from Wellington Laboratory (Guelph, Ontario). The metal solutions (lead, chromium, copper, and cadmium) were from SCP Science (Quebec, Canada).

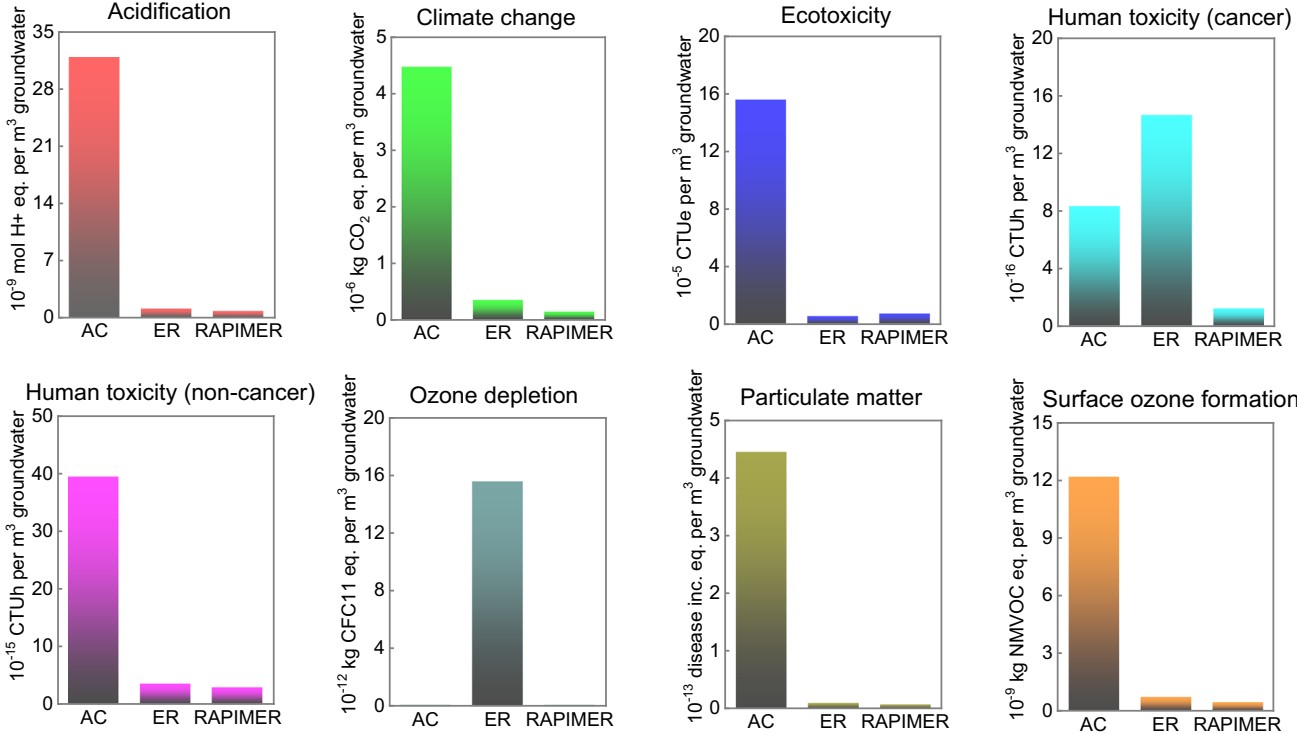

**Fig. 6 The environmental impacts of the activated carbon (AC), exchanged resin (ER) and RAPIMER for 1m³ groundwater treatment.** CTUe, comparative toxicity unit for ecotoxicity; CTUh, comparative toxicity unit for human health; CFC-11, (trichlorofluoromethane) equivalent; NMVOC, non-methane volatile organic compound.

**Lignin and cellulose separation.** An alkaline process was used to separate lignin and cellulose fibers from corn stover. In particular, 100 g of dried corn stover was added to a 2 L flask with 1000 mL sodium hydroxide solution (1% w/v) and then heated by an Amsco LG 250 Laboratory Steam Sterilizer (Steris, USA) for 1.5 h at 120 °C. After cooling to room temperature, the cellulose fibers were separated by the grade 1 waterman filter paper and then washed with distilled water until the pH was neutral. The resultant cellulose fibers were stored in the 4 °C refrigerator for further treatment. The residual lignin solution was kept for lignin recovery.

**Cellulose nanofibrils preparation.** The TEMPO-oxidization method was employed to prepare corn stover based cellulose nanofibrils. They were produced via alkaline lignocellulosic pulp processing where 4 grams of cellulose fibers were dispersed in distilled water containing 0.06 g TEMPO and 0.4 g NaBr for 30 min, and then 20 mmol NaClO solution (13%, w/w) was added. In turn, pH was controlled at 10 using the addition of 0.5 M NaOH solution and given 1 h to diffuse. The oxidized cellulose fibers were then filtered and washed with distilled water until the pH was neutral. After that, a high-pressure homogenizer was used to further process the fibers. The resultant 0.6% (w/w) cellulose nanofibrils suspension was then stored in the 4 °C refrigerator for further processing.

**Lignin recovery from the residual lignin solution.** In turn, 500 mL of residual lignin solution was first dialyzed with a 1000 MW dialysis tube. After dialysis, a HCl solution (0.5 mol/L) was added to precipitate the lignin and was allowed to work until the pH reached 2. Then the resultant liquor was centrifuged at 6000 rpm for 5 min and filtered. This yielded about 5 g of precipitated lignin. That lignin was separated, washed with acidified water at pH 2. The result was then freeze vacuum dried for about 24 h until a constant weight was observed. The final corn stover lignin samples were stored in a desiccator for further characterized or used. For modified lignin, a 500 ml of residual lignin solution with 1% lignin content was first dialyzed with a 1000mw dialysis tube for 3 days to remove extra small molecular saccharide, and then CO2 was used to adjust pH to 6.5 for the Mannich reaction. Then 40 ml PEI solution (10% w/v) and 40 ml formaldehyde (37% w/v) were gradually added into 1 L three-neck flask with 500 ml residual lignin solution under a water bath at 60 °C for 5 h. The PEI grated lignin particles were gradually precipitated during the reaction. The precipitate was filtered and then washed with distilled water until the pH was neutral. The samples of corn stover modified lignin (CSML) were freeze vacuum dried, and then stored in a desiccator for further characterization and use.

**Composite materials preparation.** To form the RAPIMER composite, citric acid (0.06 g) was added into a 250 mL beaker that contained 100 mL of the earlier prepared suspension with the cellulose nanofibrils (0.6 wt%) and this was mixed for 1 h. Then 0.6 g of modified lignin was added. The resulting mixture (that exhibited a cellulose fibril: modified lignin ratio of 1:1 w/w) was stirred for 1 h before then cooled in a 4 °C refrigerator overnight. The precooled aqueous gel was then frozen at −80 °C for 20 mins and the resulting frozen samples were freeze-dried in a lyophilizer at a condenser temperature of −50.0 °C under vacuum (0.0004 mbar) for two days. After the freezing dry process, the RAPIMER composite was subsequently cured in a vacuum oven at 80 °C for 4 h and stored in a desiccator for further use. The refrigerator cooling to the curing process was also applied to fabricate the cellulose fibers, cellulose nanofibrils, cellulose fiber/lignin composite for comparison in this study. The cellulose nanofibrils/modified lignin composition was tested at three different nanofibrils: modified lignin (w/w) ratios: 2:1, 1:1, and 1:2 to evaluate the composite biodegradability. The optimum ratio (1:1) was chosen to fabricate the RAPIMER composite.

**Contaminant adsorption experiments.** PFAS solutions were prepared for adsorption and degradation tests in batch mode using 50 mL polypropylene bottles at the pH value of 7.0 ± 0.1 (adjusted by sodium hydroxide). PFOA and PFOS solutions were prepared at various concentrations. The engineered materials were weighed, added into the PFAS solution, and then shaked in an orbit shaker at 150 rpm at room temperature. The PFAS adsorbed materials were then collected, freeze-dried, or autoclaved to prepare for further use. The HPLC-MS was used to detect the PFOA and PFOS concentration, Optical density (OD) measurement was used to measure the anionic dye concentration, and ICP-MS was used to detected Cu, Pb, Cr, and Cd concentration of the influent and effluent in the online flow testing. The element composition such as C, N, O, F, S, Na, Cl, Cu, Pb, Cr, and Cd of the RAPIMER after the adsorption test was measured by energy-dispersive spectroscopy (EDS) at an accelerating voltage of 15 kV. PFOA and PFOS solutions were prepared at various concentrations for different tests, the additional PFAS adsorption testing, PFAS quantitation and PFAS-contaminated realistic water simulation test details were described in Supplementary notes 1, 2, 3, 4, and 5.

**PFAS degradation experiments.** In preparation of the degradation experiments, the white rot fungus *Irpex lacteus (originally isolated from Shennong Nature Reserve (Hubei, China)*[46] was pre-cultured in 5 mL potato dextrose broth mounted on tissue culture plates *(VWR 10861-554)* at 28 °C for 7 days. The cultivation conditions followed procedures in the report[27]. Before inoculating onto different materials, fungal mycelium was washed three times with DI H₂O, then dispersed in 5 mL Kirk medium[26] without glucose. The engineered material

was then placed into the five mL Kirk medium without glucose on a 12 well plate to monitor the fungus growth. The as-produced composite materials, in this case, served as the sole carbon source for the fungus to grow. Kirk media with glucose was used as the control to compare the fungal growth on the engineered materials. The supernatant was filtered and tested by high-resolution LCMS analysis for PFAS degradation products.

**Material characterizations**. During the process materials were characterized at several stages to identify chemical and other characteristics. AFM analysis of cellulose nanofibrils and cellulose fibers were generated using a Bruker Dimension Icon Atomic Force Microscopy (AFM), the diameter of the fibers were analyzed by the ImageJ v1.53 s software based on AFM images. To examine the presence of chemical functional groups, Fourier Transform Infrared Spectroscopy (FTIR) spectra were generated using Thermo Nicolet 380 FTIR spectrometer in the wavelength range from 400 to 4000 $cm^{-1}$. Chemical element and chemical bonding were examined based on X-Ray photoelectron spectroscopy (XPS) spectra developed using Omicron CPS/UPS system with Argus detector and the Omicron's DAR 400 dual Al X-ray source (power of 300 W). Scanning electron microscopy (SEM) images were recorded on a Tescan FERA-3 Model GMH Focused Ion Beam Microscope at an accelerating voltage of 5 kV. Atom number and element weight ratio for major elements such as C, N, O, F, S of samples were measured by XPS and EDS. Thermogravimetric analysis (TGA) test was performed on a TA instrument TGA 5500 (TA Instruments) thermogravimetric analyzer. About 5.0 mg of sample was heated from room temperature to 700 °C at a heating rate of 10 °C under $N_2$ atmosphere. The method to calculate the density and porosity of these composite foams according to the solid density (cellulose nanofibrils density is 1.45 g/cm³ and cellulose fiber density is 1.6 g/cm³) and their volumes were previously reported[47–49]. The specific surface areas of the cellulose fibers and cellulose nanofibrils were calculated according to their carboxylic content and cationic demand[50]. All data were processed with software Origin 2021.

**Fungal viability assay**. Fungal viability was assessed by evaluating the extractable fungal proteins on different materials. A SpeedVac was used to remove the liquid solution from the fungus growing material after collecting the growing solution. 100 μL 0.5 M NaOH was then added onto the material. The resulted material solution mixture was boiled for 5 mins. After that, 100 μL 0.5 M HCl was added before centrifuging at 12,000 rpm to obtain the protein supernatant. The 50 μL supernatant was added into 1 mL Bradford solution for OD$_{595}$ measurement and construction of the standard curve to quantify the protein concentration. 1.5 μL DI water was added into 1 mL Bradford solution for comparison as control. For the Bradford protein quantitative measurement, 0.1 mg/mL, 0.2 mg/mL, 0.4 mg/mL, 0.6 mg/mL, 0.8 mg/mL and 1 mg/mL BSA solution was added into 1 mL Bradford solution, and mixed well to construct the calibration curve. For fungal viability growing on different materials, the fungal samples were collected to evaluate protein concentration on days 1, 3, 7, and 14. For the fungal viability on the PFAS enriched RAPIMER at different PFAS concentrations, the fungal samples were collected to test protein concentration on days 1, 3, 7, 10, and 14. The RAPIMER composition treated with 10 μg/L, 100 μg/L, 1000 μg/L, 10,000 μg/L of PFOA/PFOS mixture solution were employed to analyze the concentration effect of PFAS on Fungus growth. Kirk media with glucose was used as the control in the fungal viability assay.

**Extracting protein from the fungus for proteomics analysis**. The mycelia of grown on the engineered materials were collected by centrifuging at 5008 × g, washed twice with DI H$_2$O, and dried with tissue paper. 100 mg of each harvested mycelium sample was used for protein extraction. After grounding in liquid nitrogen, the mycelia powders were mixed with 1 mL Alkali-SDS buffer (5% SDS; 50 mM Tris-HCl, pH 8.5; 0.15 M NaCl; 0.1 mM EDTA; 1 mM MgCl$_2$; 50 mM Dithiothretiol) and boiled for 10 min[51]. The supernatant of the boiled sample after centrifuging at 3019×g for 10 min was mixed with 100% chilled trichloroacetic acid (TCA) with a ratio of 4:1 and then incubated at −20 °C for 2 h. Pellets containing protein samples were collected with centrifugation (15596 × g for 30 min at 4 °C) and washed twice with 1 mL chilled acetone, before air-drying and dissolving in a solution buffer contains 7 M urea, 2 M thiourea, 40 mM triszma base, and 1% 3-(4-Heptyl) phenyl-3-hydroxypropyl dimethylammoniopropanesulfonate (C7BzO). The extracted protein samples were stored at −80 °C prior to LC-MS/MS analysis.

**Protein analysis via MudPIT-based shot-gun proteomics**. The extracted proteins were analyzed with MudPIT-based shot-gun proteomics. Briefly, 100 μg of protein was first digested by Trypsin (Mass Spectrometry Grade, Promega, WI, USA) at 37 °C for 24 h, with the ratio of 1:40 w/w. A Sep-Pak Plus C18 column (Waters Limited, ON, Canada) was applied to desalt the digested peptides, which were subsequently loaded into a customized biphasic capillary column. The column, which included 5 cm of C18 reverse-phase resin (C18-AQ, The Nest Group Inc, MA, USA) and 3 cm of strong cation exchange (SCX) resin Poly-SULFOETHYL A, (The Nest Group Inc, MA, USA), was used to provide 2-dimensional separation of peptides, as described by Washburn et al[52]. The Thermo Fisher Orbitrap Velos Pro mass spectrometer (Thermo Fisher Scientific, San Jose, CA) was used for peptide detection. The mass spectrometer was set to record the peptides over a 300–1700 $m/z$ range. The five most abundant ions were then subjected to the following tandem mass (MS/MS) analysis. The Xcalibur software (Thermo Fisher Scientific, San Jose, CA) was used for LC-MS system control and data collection.

**Proteomics data analysis**. Tandem mass spectra were extracted from the raw files and converted into the MS2 file. The *Irpex lacteus* proteome and functional annotations were obtained from MycoCosm[53]. The MS2 file was searched against the filtered models which contain one representative protein per gene locus of the 15,319 gene models from the *I. lacteus* genome[53]. A ProLuCID (ProLuCID 1.3.5) algorithm with precursor tolerance of 50 ppm was used to search for data using the Texas A&M High Performance Research Computing clusters[54]. The peptide spectrum matches (PSMs) were filtered through a target-decoy strategy using a semi-supervised machine learning algorithm Percolator with false discovery rate cutoff as 0.1[55]. Spectral counts for each detected protein were computed by crux spectral counts function[56]. Differential protein expression analysis was performed by R Bioconductor package 'DEP' using the raw spectral counts[57]. Proteins expressed in at least two samples were retained. The missing values including missing at random and missing not at random were computed by 'knn' method and 'MinProb' method, respectively. Row counts of 4,876 detected proteins were normalized by variance stabilizing normalization. Significantly differentially expressed proteins were identified with adjusted $p$-value < 0.1 and fold change > 2. The normalized log2 spectral 1/3count of the proteins was used to represent their expression levels at each condition. Details of fold change and adjusted $p$-value for the differentially expressed proteins were included in Supplementary excel sheet 1. Gene ontology (GO) enrichment of differentially expressed proteins was conducted under the detected protein background using BiNGO in Cytoscape 3.9.1[58,59].

**Environmental assessment**. The multi-dimensional LCA analysis was conducted using OpenLCA 1.10.3 software (https://www.openlca.org/). A cradle-to-gate system boundary was defined. The inventory data used was primarily adopted from the Ecoinvent 3.7 database. We specified two functional units to facilitate the comparisons with alternative sorbents, (1) environmental impacts per kg of produced sorbent materials; and (2) environmental impacts in treating 1 m³ of PFAS-contaminated groundwater. See details in Supplementary note 6.

**Reporting summary**. Further information on research design is available in the Nature Research Reporting Summary linked to this article.

## Data availability

The data generated in this study are provided in the Supplementary Information/Source Data file to ensure access to the minimum dataset. The processed data sets are available as a supplementary dataset excel file to this manuscript. Source data are provided with this paper.

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

## Acknowledgements
The authors would like to acknowledge the funding support from National Institutes of Health grant R01ES032708 to S.Y.D. and J.S.Y.

## Author contributions
J.L., S.Y.D. and J.S.Y. proposed the conceptualization. J.L. designed, fabricated and characterized the materials. X.L. carried out the biological, proteomics, and the chemical analysis. J.Y. and B.L. analzyed the proteomics data. J.L., Y.D., B.A.M. performed the LCA analyses. J.L., S.Y.D., P.Z., B.L., C.B. and J.S.Y. discussed the results. J.L., X.L., B.L., Y.D., and S.Y.D. wrote the manuscript. J.S.Y. and B.A.M. edited the manuscript. S.Y.D. and J.S.Y. supervised the research.

## Competing interests
The authors (J. L. and S. Y. D.) claim competing interests. A patent is filed on the material developed in this study through Texas A&M University. There are no other competing interests.
