## [Peer Review File · Nature Communications]

Sustainable Environmental Remediation via Biomimetic Multifunctional Lignocellulosic Nano-framework Synergizing Effective BiodegradationREVIEWER COMMENTS

Reviewer #1 (Remarks to the Author):

It is suggested to make minor revisions to this submission as followed :

1. In a scientific report, please try to avoid using words like I, we, our, you, and they. We are presenting a scientific thing but not an oral presentation which can be extended with exaggeration.
2. As a kind of commercial adsorbent, the influence of mixed ions in wastewater on the adsorption capacity should be not ignored.
3. Recycling performance is an important factor to determine whether an adsorbent can be used commercially, which is not reported in this submission.
4. Mixed terminology such as adsorbent, sorbent, rpm, RCF has been used in the manuscript.
5. Please remove the typographical errors present in the manuscript.
6. The abstract needs to be revised and keywords should be below the abstract. Highlights of research are also missing.

Reviewer #2 (Remarks to the Author):

The paper is definitely well written and present a new solution to a current problem, as the presence of PFAS in wastewaters.

I have just few comments.

The authors test RAPIMER system against a controlled system. Shouldn't be validated against real effluents contaminated with PFAS and other micropollutants? Un-target chemical analysis may be challenging, but it could target the most abundant compounds. The stability and efficiency of the system against a real matrix should be addressed.

As regards the PFAS tested concentration, I suggest including a comment explaining why they were chosen. Are they representative of environmental ones?

Authors state that the metabolites produced after the fungal treatment are less toxic than the parent ones. This is true, but ecotoxicity is a phenomenon that is deeply affected by the synergic effect of many compounds together. The fact that the toxicity is reduced should be confirmed. Besides, the LCA analysis also study the ecotoxicity and the impact on human health. Do this information should be included too into the analysis? If the RAPIMER system is capable of separating PFAS from polluted waters and actively transforming them into something less toxic, this should be highlighted. Common sorbent such as activated carbon may extract pollutants from a certain matrix, but then the pollutants are still there, harmful but just in another phase. Concerning the Nagoya protocol statement, the origin of the *Irpex lacteus* strain should be indicated.

Reviewer #1 (Remarks to the Author):

"It is suggested to make minor revisions to this submission as followed: "

Response to comment: We thank the referee for the positive feedback and suggesting the publication of our manuscript after minor revisions. We have revised our manuscript to fully address the referee's comments and suggestions.

1. "In a scientific report, please try to avoid using words like I, we, our, you, and they. We are presenting a scientific thing but not an oral presentation which can be extended with exaggeration."

Response to comment: We thank the referee for pointing out this issue. We have revised our manuscript to change the active voice to the passive voice.

2. "As a kind of commercial adsorbent, the influence of mixed ions in wastewater on the adsorption capacity should be not ignored."

Response to comment: We agree with the reviewer that the influence of mixed ions in wastewater cannot be ignored. As a matter of fact, we have tested the influence of mixed ions in the original work. To simulate the realistic environmental matrices, rainwater samples were used by spiking anionic dye (direct red 81), chromium (Cr), cadmium (Cd), copper (Cu), lead (Pb), and PFOA/PFOS (50:50) solutions at the 1 µg/mL concentration. As shown in Figure 3f, the PFAS adsorption was tested in the presence of co-contaminants in the rainwater. The results highlighted that the RAPIMER could effectively adsorb PFOA and PFOS in the presence of mixed ions in simulated environmental samples (Supplementary Page 4 line 74). We also discussed the results and effects of co-contaminants in the water in the original manuscript (Page 7 and 8). The text reads as follows:

"Furthermore, RAPIMER was tested for water treatment under more realistic conditions in the presence of potential co-contaminants of PFAS. Rainwater was collected, filtered, and spiked with a mixture of PFAS, heavy metals, and another organic pollutant. Fig. 3f showed the high removal efficiency of PFAS together with five other contaminants (i.e., an anionic dye molecule, chromium (Cr), cadmium (Cd), copper (Cu), and lead (Pb) at 1mg/L for each contaminant in the rainwater) when using the same flow system (Fig. S8). The result suggested that around 99% of all the pollutants were adsorbed by the RAPIMER packed filter (Fig. 3f)."

In order to further address the reviewer's comment regards wastewater matrices, we have obtained the wastewater from the Texas A&M Wastewater Treatment Plant and tested PFOA and PFOS absorption in the wastewater effluent. The results highlighted RAPIMER can achieve similar efficiencies for PFAS absorption in wastewater, compared to the simulated wastewater (spiked rainwater), as shown in Figure 1 here. Both our original study on the simulated wastewater and this study on wastewater highlighted that RAPIMER has high specificity for PFAS and tolerance to the mixed ions' impacts. Mixed Ions in wastewater have limited impacts on RAPIMER adsorption capacity.

Figure 1. PFAS and PFOS adsorption in wastewater (experiments performed in triplicates).

3. "Recycling performance is an important factor to determine whether an adsorbent can be used commercially, which is not reported in this submission."

Response to comment: We thank the referee for this comment. Recycling performance or regeneration capacity is an important feature for conventional commercial sorbents as they are costly. However, our RAPIMER is fundamentally different from the conventional commercial adsorbent design. RAPIMER focuses on using inexpensive raw material (i.e., plant biomass) and biodegradability to enable the "treatment-train" integration, where the RAPIMER with enriched contaminants will be degraded altogether by fungus. Thus, contrary to commercial sorbents focusing on recyclability, biodegradability is an important feature for RAPIMER. The study has measured the RAPIMER biodegradability, as shown in Figure 5c. In this evaluation, RAPIMER was used as the sole carbon source to sustain fungus growth.

Due to the inexpensive feedstock and the unique 'treatment-train' integration design, RAPIMER is not set to recycle yet to be degraded together with the contaminants, to achieve the environmental benefits and avoid the often-expensive recycling process. Overall, the design of RAPIMER adsorbent from natural biopolymers overcomes the downstream challenges in contaminant treatments, enables RAPIMER serves as a substrate for microbial biodegradation while co-metabolizing the PFAS contaminants. This PFAS/adsorbent/microbe co-metabolization concept is different from conventional commercial sorbents requiring recycling, which is also a strength of the RAPIMER design.

4. "Mixed terminology such as adsorbent, sorbent, rpm, RCF has been used in the manuscript."

Response to comment:

Response to comment: We thank the reviewer for pointing out the inconsistency. We revised the manuscript and unified the terminology. We replaced "adsorbent" with "sorbent". We also converted all reactive centrifugal force (RCF) units to revolutions per minute (rpm) (manuscript pages 17-18). The edited text now reads as follows:

"Extracting protein from the Fungus for proteomics analysis. The mycelia of the white rot fungus grown on the engineered materials were collected by centrifuging at 8500 rpm, washed twice with DI H₂O, and briefly dried with tissue paper. 100 mg of the harvested mycelium sample was then grounded in liquid nitrogen and boiled for 10 min in 1mL Alkali-SDS buffer (5% SDS; 50 mM Tris-HCl, pH 8.5; 0.15 M NaCl; 0.1 mM EDTA; 1 mM MgCl₂; 50 mM Dithiothreitol)⁵⁰. The supernatant of the boiled sample after centrifuging at 6600 rpm for 10 min was transferred to a fresh tube. To each tube, 100% chilled trichloroacetic acid (TCA) was added to a final concentration of 20%. The solution was mixed well and incubated at -20 °C for 2 hours. Samples were centrifuged at 15000 rpm for 30 min at 4 °C to remove the supernatant. The pellet was harvested and washed twice with 1 mL chilled acetone following centrifuging at 15000 rpm for 30 min at 4 °C. The protein pellet was air-dried and then dissolved in a solution buffer contains 7 M urea, 2 M thiourea, 40 mM triszma base, and 1% 3-(4-Heptyl) phenyl-3-hydroxypropyl dimethylammoniopropanesulfonate (C7BzO). The extracted protein pellet was stored at -80°C prior to LC-MS/MS analysis."

5. "Please remove the typographical errors present in the manuscript."

Response to comment: Thank you very much for this comment. We carefully revised this manuscript and corrected the typographical errors.

6. "The abstract needs to be revised and keywords should be below the abstract. Highlights of research are also missing."

Response to comment: Thank you very much for this comment. We revised the abstract based on the reviewer's suggestions and added keywords. The abstract now reads as follows:

*"Chemical pollution threatens human health and ecosystem sustainability. Persistent organic pollutants (POPs) like per- and polyfluoroalkyl substances (PFAS) are expensive to clean up once emitted. Innovative and synergistic strategies are urgently needed, yet process integration and cost-effectiveness remain challenging. An in-situ PFAS remediation system was developed to employ a plant-derived biomimetic nano-framework to achieve highly efficient adsorption and subsequent fungal biotransformation synergistically. The multiple component framework is presented as Renewable Artificial Plant for In-situ Microbial Environmental Remediation (RAPIMER). RAPIMER exhibits record adsorption capacity for the PFAS compounds and diverse adsorption capability toward co-contaminants. Subsequently, RAPIMER provides the substrates and contaminants for in situ bioremediation via fungus *Irpex lacteus* and promotes PFAS detoxification. RAPIMER arises from cheap lignocellulosic sources, enabling broader impact on sustainability and a new means for low-cost pollutant remediation."*

Keywords: Chemical pollution, PFAS, bioremediation, fungal remediation, biomimetic material

Reviewer #2 (Remarks to the Author):

"The paper is definitely well written and present a new solution to a current problem, as the presence of PFAS in wastewaters."

Response to comment: We are grateful for the comments. We are delighted that the reviewer shares our excitement about novel techniques for PFAS treatment. We have carefully taken into account the comments by the reviewer and addressed them as follows.

1. "I have just few comments. The authors test RAPIMER system against a controlled system. Shouldn't be validated against real effluents contaminated with PFAS and other micropollutants? Un-target chemical analysis may be challenging, but it could target the most abundant compounds. The stability and efficiency of the system against a real matrix should be addressed."

Response to comment: We agreed with the reviewer that the real effluents containing PFAS need be validated. Therefore, to simulate the realistic contaminated wastewater with other micropollutants, we collected rainwater at College Station, Texas, and simulate the wastewater effluent with anionic dye (direct red 81), chromium (Cr), cadmium (Cd), copper (Cu), lead (Pb) and PFAS at microgram per liter level (Supplementary Page 4 line 74). The result suggested that around 99% of all micropollutants, including PFAS, were adsorbed by the RAPIMER packed filter system. We also discussed the results and effects of co-contaminants in the water on PFAS absorption in the manuscript (presented in our responses to reviewer 1). Untargeted chemical analysis is always desired regards to monitoring the micropollutants in the water samples when the broad scope surveillance is needed and the monitoring methodologies are well established. In the manuscript, we have carried out comprehensive quantitative analyses to evaluate the ions, dye, and PFAS removal. However, we feel untargeted qualitative testing probably renders less important information to evaluate the adsorption capacity when developing a new adsorbent. Quantitative measurements to accurately evaluate the sorbents capacity and co-contaminant removal are thus used as the primary evaluation tool in the current study.

Furthermore, to address the reviewer's comment on 'the stability and efficiency of the system against a real matrix', we carried out the PFOA and PFOS absorption on RAPIMER in natural wastewater collected from the Texas A&M Wastewater Treatment. The result is presented in the responses to Reviewer 1's comment No. 2 and Figure 1 of this document.

2. "As regards the PFAS tested concentration, I suggest including a comment explaining why they were chosen. Are they representative of environmental ones?"

Response to comment: We agreed with the reviewer that we need to include a comment explaining for PFAS tested concentration. We addressed the comment by expanding the information in the manuscript and supplementary document. To fully understand the absorption performance and mechanism, the low PFAS concentration (1 µg/L) is chosen to simulate the realistic condition of wastewater (See supplementary note 4). We thus edited the original manuscript by adding one reference that describes the PFAS concentrations in water and wastewater (Page 7, lines 183-187 and the new reference 32). The text now reads "The RAPIMER material was packed into an online filter in a flow system (Fig. S8) and subject to filtering a PFAS solution at 1 µg/L concentration. The concentration is lower than many of the environmental relevant conditions, where PFAS are detected up to 20 µg/L or even higher concentrations³². The lower concentration is to sufficiently test the absorption capacity at low concentration."

Meanwhile, the high PFAS tested concentration (at the mg/L concentration range) has been used to compare different sorbents for PFOA and PFOS removal rates from literature (See Table S3, measurement described in Supplementary note 2 and 3). The selection of this wide range of concentrations thus highlights the capacity of RAPIMER in PFAS adsorption and removal.

3. "Authors state that the metabolites produced after the fungal treatment are less toxic than the parent ones. This is true, but ecotoxicity is a phenomenon that is deeply affected by the synergic effect of many compounds together. The fact that the toxicity is reduced should be confirmed. Besides, the LCA analysis also study the ecotoxicity and the impact on human health. Do this information should be included too into the analysis? If the RAPIMER system is capable of separating PFAS from polluted waters and actively transforming them into something less toxic, this should be highlighted. Common sorbent such as activated carbon may extract pollutants from a certain matrix, but then the pollutants are still there, harmful but just in another phase.

Response to comment: We agreed with the referee's comment that "ecotoxicity is a phenomenon that is deeply affected by the synergic effect of many compounds together." It is inappropriate to simply assume the metabolites are less toxic than the parent molecules. However, extensive research established that the shorter chains C4, C5, and C6 are less toxic than the C8 PFAS. We have cited the reference by Mahapatra, C. T. et al. in our original manuscript. Mahapatra, C. T. et al. demonstrated that "PFASs with shorter

carbon chains (PFHA (C6) or PFBA(C4)) are less toxic than PFOA." Our study has detected the potential metabolites of perfluorohexanoic acid, which is the exact metabolite studied by Mahapatra, C. T. et al. Furthermore, a recent 2022 study (Palazzolo, S., et al., 2022, "Early Warnings by Liver Organoids on Short-and Long-Chain PFAS Toxicity". *Toxics*, 10(2), 91.) has also supported that PFBA (C4) and PHPA (C5) are potentially less toxic than the PFOA (C8) using totally different technology than the Mahapatra, C. T. et al. study. Thus, we feel that studying the degraded PFAS metabolites and their toxicity is counter-productive and beyond the scope of the current study. The Palazzolo study has been cited as THE NEW reference 37. To clarify the point, we have expanded on the discussion of the ecotoxicity of different PFAS compounds, citing the new reference 37. We thus expanded our discussion as follows (Page 10, line 284-287):

"Previous studies by Mahapatra et al. and others have shown that the shorter chain PFBA (C4), PHPA (C5), and PFHA (C6) are much less toxic than the PFOA (C8), suggesting the degraded products detected in this study are potentially less toxic than PFOS and PFOA^{30,37}"

At the same time, we invite the reviewer and the editor to consider the ecotoxicity modeling in the LCA model. The LCA analysis is based on the raw materials used to manufacture the adsorbent and the ecotoxicity caused by the manufacturing process. The current LCA analysis did not include the treated chemicals (PFAS) themselves. Thus, the additional benefit of degrading PFAS is not included in the current LCA analysis. We agree with the referee that this additional benefit should be highlighted, which was presented in the original submission on page 12, line 339 to 343, in the revised version on page 12, line 348-352.

The original discussion reads as follows:

"RAPIMER degradability, PFAS degradation, and lack of secondary pollutions lead to additional environmental benefits that were not included in this LCA analysis⁴⁴. Use of less energy, and no significant PFAS reemission are also benefits relative to existing practices⁴⁵. In addition, its self-degradation alleviates the need for additional equipment and additional associated environmental burdens."

In summary, to highlight the advantage of RAPIMER over other sorbent materials (i.e., activated carbon), we performed the LCA analysis using input data from the literature for each sorbent material. The comparative LCA analysis suggested that the RAPIMER system is indeed less toxic to the environment, especially in the category of freshwater ecotoxicity.

4. "Concerning the Nagoya protocol statement, the origin of the *Irpex lacteus* strain should be indicated." Response to comment: The origin of the *Irpex lacteus* strain information and related reference (reference 46) is provided (page 16, line 452-455) in the manuscript.

The edited text reads as follows:

*"In preparation of the degradation experiments, the white rot fungus *Irpex lacteus* (originally isolated from Shennong Nature Reserve (Hubei, China)⁴⁶ was pre-cultured in 5 mL potato dextrose broth mounted on tissue culture plates (VWR 10861-554) at 28 °C for 7 days."*

Again, we appreciate the suggestions and comments from the reviewers and this opportunity to revise our manuscript. We hope the revised version has addressed all the comments and concerns. We look forward to hearing back from you.

Sincerely yours,

Susie Y. Dai

Texas A&M University

REVIEWERS' COMMENTS

Reviewer #1 (Remarks to the Author):

The authors have now adequately addressed the concern and the paper may be accepted for the publication.

Reviewer #2 (Remarks to the Author):

I thank the authors for the revisions made. I do not have any more comments.